# Atlantic Salmon Mucins Inhibit *LuxS*-Dependent *A. Salmonicida* AI-2 Quorum Sensing in an *N*-Acetylneuraminic Acid-Dependent Manner

**DOI:** 10.3390/ijms23084326

**Published:** 2022-04-13

**Authors:** János Tamás Padra, Stefany Ojaimi Loibman, Kaisa Thorell, Henrik Sundh, Kristina Sundell, Sara K. Lindén

**Affiliations:** 1Department of Medical Chemistry and Cell Biology, University of Gothenburg, 405 30 Gothenburg, Sweden; janos.tamas.padra@gu.se (J.T.P.); stefany.loibman@gu.se (S.O.L.); kaisa.thorell@gu.se (K.T.); 2Department of Biological and Environmental Sciences, University of Gothenburg, 405 30 Gothenburg, Sweden; henrik.sundh@bioenv.gu.se (H.S.); kristina.sundell@bioenv.gu.se (K.S.)

**Keywords:** sialic acid, signaling, bacterial growth, biofilm, mucus

## Abstract

One of the most important bacterial diseases in salmonid aquaculture is furunculosis, caused by *Aeromonas salmonicida*. Bacterial communication through secreted autoinducer signals, quorum sensing, takes part in the regulation of gene expression in bacteria, influencing growth and virulence. The skin and mucosal surfaces, covered by a mucus layer, are the first point of contact between fish and bacteria. Mucins are highly glycosylated and are the main components of mucus. Here, we validate the *Vibrio harveyi* BB170 bioreporter assay for quantifying *A. salmonicida* quorum sensing and study the effects of Atlantic salmon mucins as well as mono- and disaccharides on the AI-2 levels of *A. salmonicida*. Atlantic salmon mucins from skin, pyloric ceca, proximal and distal intestine reduced *A. salmonicida* AI-2 levels. Among the saccharides abundant on mucins, fucose, *N*-acetylneuraminic acid and GlcNAcβ1-3Gal inhibited AI-2 *A. salmonicida* secretion. Removal of *N*-acetylneuraminic acid, which is the most abundant terminal residue on mucin glycans on Atlantic salmon mucins, attenuated the inhibitory effects on AI-2 levels of *A. salmonicida.* Deletion of *A. salmonicida* *luxS* abolished AI-2 production. In conclusion, Atlantic salmon mucins regulate *A. salmonicida* quorum sensing in a *luxS* and *N*-acetylneuraminic acid-dependent manner.

## 1. Introduction

Atlantic salmon (*Salmo salar* L.) aquaculture is a large and growing food industry. In aquaculture worldwide, disease outbreaks are a limiting factor for ethical and economically sustainable production. One of the most severe diseases in fish farming is furunculosis, caused by *Aeromonas salmonicida* ssp. *salmonicida*, which primarily affects the members of the *salmonidae* family [1,2]. A thorough understanding of host–pathogen interactions can enable novel ways of treating infections.

Fish are covered by a continuously secreted mucus layer, which is the first barrier that pathogens encounter [3]. The main components of this mucus layer are secreted mucins. We have shown that Atlantic salmon mucins carry over a hundred different carbohydrate structures, and thus provide a vast array of potential binding sites for microbes [4,5] The mucins therefore constitute an important part of the mucosal defense against infections [6,7]. The interaction between Atlantic salmon mucins and *A. salmonicida*, i.e., adhesion and growth of pathogens in response to mucins, differs depending on which organ of the fish the mucins were isolated from, is governed by the mucin glycans, and is influenced by environmental factors [5,8]. 

Bacteria form dynamic communities with distinct social behaviors. Communication between microbes occurs by means of small extracellular molecules, also called autoinducers, and is referred to as quorum sensing (QS) [9]. *A. salmonicida* produces *N*-acylhomoserine lactone (AHL) molecules [10], which are characteristic of numerous Gram-negative bacteria. AHL molecules are termed autoinducer-1 molecules (AI-1). Another major signaling molecule utilized by *A. salmonicida* is the autoinducer-2 molecule (AI-2) [11,12] which is the derivative of the 4,5-dihydroxy-2,3-pentanedione (DPD) molecule [13]. The latter system can be found in members of both Gram-negative and Gram-positive bacteria and is the “common language” of interspecies communication [14]. In *A. salmonicida,* S-ribosylhomocysteine lyase encoded by the *luxS* gene synthesize AI-2 [15]. The *luxS* mutant *A. salmonicida* was shown to overexpress two virulence factors: *plc* (phospholipase C) and *ahe2* (serine protease) but underexpressed *vapA*, that codes for the A-layer protein and is a determining factor in cell morphology. Moreover, *luxS* mutant *A. salmonicida* shows lower levels of autoaggregation and triggers a more rapid immune response in Atlantic salmon compared to the wild-type strain [12]. *A. salmonicida* and its close relative *A. hydrophila* possess a range of virulence factors, including α-hemolysin [16], glycero-phospholipid-cholesterol acyltransferase [17,18], lipase [19] and serine protease [10,20]. The expression of several of these exoproducts is associated with high cell densities in the late exponential/stationary phase [10,16,17,18,19,20]. *A. salmonicida* QS is likely to take an important part in the regulation of bacterial-cell density and virulence during the pathomechanism of infection. AI-2 signaling is less explored compared to the AI-1 system, and its role needs to be studied further in *A. salmonicida*. 

A widely used reporter organism for QS is *V. harveyi*. The cell density and luminescence of *V. harveyi* is controlled by various QS signals. *V. harveyi* BB170 sensor I^¯^ reporter strain (hereafter called *V. harveyi* BB170) lacks the *luxN* sensor domain [21] and therefore lacks the phosphatase activity that can remove phosphate from LuxO to induce bioluminescence expression [22] upon contact with AI-1 signals. Thus, *V. harveyi* BB170 does not respond to AI-1 signals, making it suitable for the specific detection of AI-2 signals.

The main objectives of the project were to validate the *V. harveyi* BB170 bioreporter assay in *A. salmonicida* QS and to study the effects of Atlantic salmon mucins on the AI-2 levels of *A. salmonicida*. By creating a *luxS* mutant and using mono- and disaccharides present terminally on the mucin glycans, we further elucidated the mechanisms behind the inhibitory effects of mucins on AI-2 signaling.

## 2. Results

### 2.1. V. Harveyi BB170 Autoinducer-2 Reporter Assay Is Suitable for AI-2 Quantification in Narrow Concentration Ranges Defined by the Selected Culture Medium

We first tested the linear range of the AI-2 reporter assay by using the supernatant of *V. harveyi* BB170 (Figure 1A). The detection was linear between the undiluted supernatant and the four-fold dilution, but not with further dilutions. Using a dilution series of 4,5-dihydroxy-2,3-pentanedione (DPD) ranging from 100 µM to 1 nM, we found that the detection range in M9 minimal medium supplied with casamino acids (CM9 medium) was 10–100 nM, whereas in defined medium (DM) the range was 100–1000 nM (Figure 1B). The relationship between DPD concentration and relative luminescence was linear, reflecting a good fit in the aforementioned ranges (CM9 r^2^ = 0.98 and DM r^2^ = 0.98). Below and above these ranges, the assay did not allow quantification of AI-2 signals due to lack of linearity in the relationship between *V. harveyi BB170* luminescence and DPD concentration (Figure 1B). Two reasons that might explain the differences in detection range between these media are differences in opacity that affect the luminescence signal, and also differences in nutrient compositions that affect the behavior of *V. harveyi* in the reporter assay. 

### 2.2. A. Salmonicida Autoinducer-2 Production Starts before the Log Phase of Growth and Its Kinetics Differs Depending on Culture Media

The timecourse of extracellular AI-2 level changes differed depending on if *A. salmonicida* was cultured in CM9 (Figure 2A) or DM (Figure 2B). The common trait was that the majority of the signals were produced before the log phase started. The AI-2 increase in CM9 was slow, reaching maximum levels after 9 h (Figure 2A) compared to 2 h in DM medium (Figure 2B). In CM9, the largest increase occurred before the log phase and the increase continued at a slower rate (Figure 2A). In DM, the level of AI-2 reached a plateau in the lag phase and their level remained similar throughout the time course (Figure 2B).

### 2.3. Exogenous DPD Enhances A. Salmonicida Growth

*A. salmonicida* responded to DPD treatment with increased growth in both media (Figure 3A,B). In CM9, OD_560_ of the cultures increased linearly with increasing DPD concentration (ΔOD_560_ in log phase vs. DPD conc.: r^2^ = 0.91; *p* = 0.0002; *n* = 4, Figure 3A). In DM, the increase in OD_560_ in response to DPD treatment was less pronounced (ΔOD_560_ in log phase vs. DPD conc.: r^2^ = 0.66; *p* = 0.0078, Figure 3B; *n* = 4). CFU/mL vs. DPD growth curves of *A. salmonicida* calculated using CFU/mL-OD_560_ standard curves in the logarithmic range of growth (Figure 3C,D) indicate a stronger effect of DPD on growth in CM9 compared to DM. *A. salmonicida* doubling times calculated from the logarithmic growth phase inversely correlated with DPD concentrations in both media (r^2^ = 0.82; *p* = 0.002, *n* = 4 in CM9, Figure 3E and r^2^ = 0.58; *p* = 0.0172, *n* = 4 in DM, Figure 3F).

### 2.4. Atlantic Salmon Mucins Reduce A. Salmonicida AI-2 Secretion

Atlantic salmon mucins from skin, pyloric ceca, proximal and distal intestine reduced AI-2 signals from *A. salmonicida* cultures (*p* = 0.0001, *p* = 0.0049, *p* = 0.0001 and *p* = 0.0008, respectively, *n* = 5, Figure 4A). Skin mucins reduced AI-2 secretion by 81%, proximal intestinal mucins by 72%, distal intestinal mucins by 56% and pyloric cecal mucins by 48%. The effect of mucins on the AI-2 production and *A. salmonicida* growth did not correlate (r^2^ = 0.05, *p* = *n*.s., *n* = 19; Figure 4B), which together with the results in Figure 2 suggest that mucin effects on *A. salmonicida* growth are not the main cause for the effects on *A. salmonicida* AI-2 production.

### 2.5. Among the Saccharides Abundant on Mucins, Fucose, N-Acetylneuraminic Acid and GlcNAcβ1-3Gal Inhibit AI-2 Secretion of A. Salmonicida

The disaccharide GlcNAcβ1-3Gal, which is a common epitope on Atlantic salmon mucins, reduced the AI-2 levels in the *A. salmonicida* supernatant by 31% (*n* = 6; *p* = 0.0297; Figure 5). Fucose decreased AI-2 levels by 44% (*n* = 6; *p* = 0.0008; Figure 5) while NeuAc decreased them by 35% (*n* = 6; *p* = 0.0101; Figure 5). Galactose, GalNAc and the galactose-containing saccharides lactose and GalNAcβ1-3Gal also had a tendency to reduce AI-2 production in *A. salmonicida* cultures in DM (*n* = 6; *p* = *n*.s.; Figure 5), whereas no such tendency was present for *N*-glycolylneuraminic acid and *N*-acetylglucosamine.

### 2.6. Removal of N-Acetylneuraminic Acid from Atlantic Salmon Mucins Attenuates the AI-2 Reducing Effects of Mucins on A. Salmonicida

Among the terminal monosaccharides located on salmon mucins that reduced *A. salmonicida* AI-2 production (Figure 5), *N*-acetylneuraminic acid is the most prevalent [8]. Enzymatic removal of *N*-acetylneuraminic acid from salmon mucins increased the AI-2 production of *A. salmonicida* (skin, pyloric ceca, proximal intestine, and distal intestine *p* = 0.0041, *p* = 0.0021, *p* = 0416 and *p* = 0.0035, respectively, Figure 6). *N*-acetylneuraminic acid removal from both skin and pyloric cecal mucins increased the AI-2 levels by 32% compared to mock-treated mucins (Figure 6A,B), whereas removal from proximal and distal intestinal mucins increased AI-2 levels by 22% and 25%, respectively (Figure 6C,D).

### 2.7. Identification of the luxS Gene in the Genome of A. Salmonicida Ssp. Salmonicida Strain VI-88/09/03175

The genome sequence of the *A. salmonicida* ssp. *salmonicida* strain VI-88/09/03175 was determined, resulting in a draft genome of 4,738,132 bp over 123 contigs, with an average coverage of 52-fold coverage. From the annotation, a luxS homolog was identified. The luxS gene sequence of *A salmonicida* ssp. *salmonicida* strain VI-88/09/03175 (510 bp) showed a 100% homology with its homolog in the *A. salmonicida subsp. salmonicida* A449 reference genome (accession number CP000644.1). The draft genome was submitted to GenBank under BioProject number PRJNA698804.

### 2.8. Construction of ΔluxS Gene-Deletion Mutant Strain

To investigate the relationship between luxS gene and AI-2 activity in *A. salmonicida* we first deleted the luxS gene, creating a mutant strain by homologous recombination. As shown in Figure 7, the deletion was then verified by PCR using genomic DNA (gDNA) as a template with two approaches: (1) Amplification of the flanking regions of luxS gene with luxS-up-F and luxS-down-R primers produced bands with 980 bp for the wild type (WT) while the ΔluxS mutant clones produced 470 bp bands, in accordance with the theoretical size (Figure 7B); (2) Amplification mapping inside the luxS gene with LuxS-F and LuxS-R primers produced a product only in the WT strain (Figure 7C).

### 2.9. Growth Characteristics of A. Salmonicida WT and ΔluxS

To investigate the influence of luxS on *A. salmonicida* growth, growth rates from the ΔluxS mutant and WT were compared (Figure 8). The growth for the first few hours after starting a liquid culture in fresh DM was similar for the WT and ΔluxS mutant. At later time points, the ΔluxS mutant growth lagged behind that of the WT, and by 8 h the ΔluxS mutant growth was approximately 2-fold less than that of the WT (Figure 8).

### 2.10. Deletion of LuxS Abolished A. Salmonicida AI-2 Production Both in the Absence and Presence of Skin Mucins

After 8 h of culture in DM with a starting OD_600_ of 0.1, the AI-2 production of the ΔluxS mutant was significantly decreased compared to that of WT *A. salmonicida*, as measured using the *V. harveyi* BB170 strain bioluminescence assay (Figure 9A). To further investigate the effect of Atlantic salmon skin mucins on AI-2 secretion, *A. salmonicida* WT and ΔluxS mutant strains were cultured in DM with and without Atlantic salmon skin mucins. In order to investigate if the difference in cell density caused by differences in growth rate between the ΔluxS and WT *A. salmonicida* interfered with the assay, we measured AI-2 activity after 6 and 8 h and included cultures with a starting OD_600_ of 0.4 starting for the ΔluxS in addition to the standard starting OD_600_ of 0.1 (Figure 9B). Skin mucins notably decreased the levels of secreted AI-2 molecules from the *A. salmonicida* WT strain after both 6 h and 8 h (Figure 9C). The levels of secreted AI-2 molecules from *A. salmonicida* ΔluxS were also statistically significantly decreased in the majority of cultures with skin mucins compared to those without skin mucins (Figure 9C). However, the amplitudes of both the AI-2 activity in *A. salmonicida* ΔluxS cultures without mucins and the level of inhibition in the ΔluxS cultures with skin mucins were very low (Figure 9C) and are thus less likely to have a biological impact than in the WT strain.

## 3. Discussion

In this study, we demonstrated that AI-2 production of *A. salmonicida* could be quantified with the *V. harveyi* BB170 bioluminescent-reporter system. The assay differed in sensitivity and linear range for the two *A. salmonicida* culture media used, demonstrating the necessity to investigate the linear range for different media before performing such assays. Exogenous DPD increased the growth of *A. salmonicida,* and an isogenic *luxS* deletion strain grew slower than the WT strain. Deletion of *luxS* abolished the vast majority of the *A. salmonicida* AI-2 secretion. Furthermore, Atlantic salmon mucins from the skin, pyloric ceca, proximal intestine and distal intestine all reduced the *A. salmonicida* AI-2 secretion. Out of the monosaccharides and disaccharides that we previously identified among the Atlantic salmon mucin glycan structures [4], fucose, NeuAc and GlcNAcβ1-3Gal reduced the AI-2 secretion of *A. salmonicida*. After the enzymatic removal of NeuAc, which is the most abundant terminal monosaccharide on mucins, from mucins the AI-2 secretion increased, suggesting an important role for NeuAc in the inhibition of *A. salmonicida* AI-2 signaling.

Concerns have previously been raised about the precision and applicability of the *V. harveyi* BB170 bioluminescent-reporter system in detecting AI-2 signals [23,24] claiming, for example, that the luminescence response, working as an ON and OFF switch, is only suitable for qualitative assessments, i.e., to confirm the presence or absence of AI-2. It is indeed important for assays such as the *V. harveyi* reporter assay, where the detection occurs by a living organism, to be approached with caution. During the optimization of the reporter assay, we used the reporter strain’s own cell-free culture supernatant and also *A. salmonicida* cell-free culture supernatant in a serial dilution to examine the relationship between the dilution and relative luminescence of the reporter. In both cases, the relationship was linear but in a limited window. In the detection of the commercially available AI-2 molecule, DPD, by the reporter assay, we found differences between the two media we used. It has been shown previously that nutrients in the culture medium, such as Fe (III) and borate, affect the luminescent signals of the autoinducer assay [24], which explains the difference we have seen. Vilchez et al. did not find a linear relationship between DPD concentration and luminescence [24], but their study did not include the range where we found the assay to be linear. Wang et al. found the linear range for DPD detection in aqueous solution with *V. harveyi* BB170 to be between 80 nM and 1 µM, which is similar to what we found in the DM medium [25]. Similarly, *V. harveyi* MM77 AI-1−, AI-2− had a linear response to DPD between 0.0002 and 200 nM [26]. These results suggest that the *V. harveyi* BB170 reporter system can be used for comparing AI-2 levels between treatments of bacteria cultured in the same medium. Using DPD for standard curves when comparing treatments of bacteria cultured in different media also appears to be a valid approach as long as potential differences in detectability are considered.

We found that addition of exogenous DPD in high concentrations increased *A. salmonicida* growth. The concentrations that increased growth in the present paper exceed concentrations we measured in *A. salmonicida* cultures up to ca. 200-fold, yet they can be relevant in the microenvironment of bacteria, especially in a biofilm and/or in presence of other bacteria that secrete more AI-2. Furthermore, the *A. salmonicida* VI-88/09/03175Δ*luxS* mutant that we made consistently grew slower than its WT parent strain, both in the absence and presence of mucins. The *luxS* gene is highly conserved among many different bacterial species [27,28]. In several bacterial species, including *Edwardsiella piscicida, Streptococcus pyogenes*, and *Bacillus anthracis*, deletion of *luxS* lead to decreased growth [29,30,31]. Nevertheless, no clear effect on growth was detected neither in an *A. salmonicida luxS* mutant (C4) nor in an *Aeromonas hydrophila luxS* mutant [12,32,33]. When growth effects were detected in *luxS* deletion mutants, they were not very prominent, and *LuxS*-related growth deficiencies may also be dependent on the media used in the study [30]. Taken together, these results indicate that AI-2 synthesis is not essential for growth, but *luxS* can be involved in different metabolic pathways [34]. It has been shown recently that LuxS in *A. salmonicida* has a role in regulating *vapA* expression [12]. The *luxS* mutant *A. salmonicida* has decreased expression of VapA, and consequently the cells show reduced autoaggregation [12]. Exogenous AI-2 signals may thus increase aggregation and support the growth of *A. salmonicida*, as we have seen in our experiments, and this likely happens along with increased VapA production. Increased aggregation can lead to larger errors in optical-density measurements of bacteria [35]. Therefore, we confirmed both the growth-promoting effect of DPD with alamarBlue reduction measurements (data not shown). We also confirmed the differences in growth between our *luxS* deletion mutant and its parent strain with the alamarBlue assay (Figure 7), but the outcome of OD and alamarBlue measurements were similar. Further, in response to bacteria or aerolysin-induced tight-junction disruption, molecules resembling AI-2 can be secreted by host epithelia. This phenomenon has been reported in the mammalian gut, which produced molecules mimicking AI-2 upon contact with bacteria, which led to subsequent bacterial gene expression changes [36]. The authors argued for a role of the AI-2 mimic in the symbiotic relationship with gut commensals that participate in the healing process of the epithelia. No similar phenomenon has been described in Atlantic salmon to date. However, in an Atlantic salmon-disease challenge experiment the expression levels of immune factors were more pronounced against the *luxS* deletion mutant than against wild-type *A. salmonicida* [12] suggesting a role of AI-2 for *A. salmonicida* survival in the host. It is likely that even though the intrinsic *A. salmonicida* AI-2 secretion is low, signal hijacking, i.e., sensing the common AI-2 signals produced by other bacteria, in the host can contribute to increased growth and survival of *A. salmonicida*.

In our study, the AI-2 activity in spent media from the Δ*luxS* mutant strain was significantly decreased compared to its parent *A. salmonicida* wt, suggesting that the *luxS* gene is closely related to AI-2 production. However, a small residual AI-2 activity still remained. The signal was close to the detection level of the assay, but it can not be excluded that other genes can produce molecules that elicit a low level of AI-2 response 

In the present study we characterized, for the first time, the effect of Atlantic salmon mucins on the AI-2 secretion of *A. salmonicida*. Addition of purified mucins from skin, pyloric ceca, proximal intestine and distal intestine decreased the levels of secreted AI-2 molecules in the culture supernatant of *A. salmonicida*. We found no correlation between growth rate and AI-2 levels after culturing *A. salmonicida* with mucins. This supports the claim that mucins modulate AI-2 signaling in *A. salmonicida* directly and not only as a side effect of changes in growth. Our results on the AI-2 time course of *A. salmonicida* also corroborate this statement, as we did not find AI-2 levels to tightly follow that of growth. Mucins and bacterial AI-2 signals encourage a planktonic state of the human gastric pathogen *Helicobacter pylori* over biofilm formation. This is proposed as a means of dissemination of bacteria after a certain bacterial density is reached [37,38]. This might primarily be beneficial for the bacteria, allowing it to leave a suboptimal environment in search for other targets. *LuxS* mutant *H. pylori* without AI-2 secretion formed biofilms two-fold more efficiently than wild-type bacteria [37], supporting the role of AI-2 signaling in biofilm dispersal and virulence towards the host. After coming in contact with mucins, *Vibrio cholerae* lose flagella and repress quorum sensing through the LuxO regulator, which initiates virulent behavior [39]. *A. salmonicida* does not normally have flagella [40], but their pili take part in host colonization [41,42,43]. Similarly to *V. cholerae* losing their flagella while passing through the mucus layer [39] *A. salmonicida* was found to use pili to initiate contact with the host, after which they were suggested as unnecessary in the invasion process [43]. Due to the regulatory role of QS systems in pili formation and motility [28,44] our results with reduced AI-2 secretion of *A. salmonicida* upon contact with mucins might indicate a response where the pili formation is impeded. Exogenous AI-2 increased the expression of a hemolysin gene, *aerB,* in *luxS* mutant *A. salmonicida* [12], which suggests a lower *aerB* expression in the presence of mucins. *A. salmonicida* is known to operate with a range of AI-1 molecules [10,45] and the AI-2 system. As there is a high level of interaction between these systems in bacteria, it cannot be concluded with certainty that decreased AI-2 production upon contact with salmon mucins alone leads to decreased virulence in *A. salmonicida*. Despite the knowledge gap in the QS-signaling mechanisms in *A. salmonicida*, it is certain that this pathogen not only reacts to mucins by attachment [5] and altered growth [8] but also by altering group behaviors. Taken together, this suggests both a dynamic interaction between *A. salmonicida* and salmon mucins, and that mucins modulate pathogen behavior in several ways.

To pinpoint saccharides in the mucin *O*-glycans of Atlantic salmon that can contribute to the AI-2 inhibition in *A. salmonicida*, we cultured *A. salmonicida* in the presence of mono- and disaccharides. The higher concentration of accessible carbohydrate in the CM9 medium masks the effect of the saccharides, which makes it a suboptimal medium for studying the effect of saccharides on AI-2 secretion in bacteria. Thus, we decided to study the saccharides effect in DM medium only. All studied saccharides except GlcNAc and NeuGc tended to reduce the AI-2 secretion of *A. salmonicida*, while fucose, *N*-acetylneuraminic acid and GlcNAcβ1,3GalNAc displayed a significant inhibitory effect. We have previously shown that the disaccharide GlcNAcβ1,3GalNAc increases the growth of *A. salmonicida* through GlcNAc [8]. As we normalized the AI-2 changes for growth, an alternative explanation for the inhibitory effect of this molecule and GlcNAc could be through a growth-promoting effect without changes in AI-2 levels. Other saccharides that inhibited AI-2 secretion in *A. salmonicida* in DM medium were fucose and NeuAc, which had no effect on the growth of *A. salmonicida* [8]. These two monosaccharides are present on mucins as terminal structures, i.e., structures that first come into contact with bacteria. Fucose is scarce as a terminal monosaccharide, while NeuAc is the dominant terminal monosaccharide on mucins from all investigated Atlantic salmon epithelial sites [4,8]. Interestingly, the structurally closely related NeuGc, present on skin mucins but almost entirely absent in the gastrointestinal tract, had no effect on QS. To provide further support for NeuAc as a central player in *A. salmonicida* AI-2 signaling, we treated Atlantic salmon mucins with the sialidase A enzyme that cleaves most NeuAc from the terminal positions of the *O*-glycans, as previously validated [5]. Sialidase treatment of mucins from all four epithelial sites increased AI-2 levels in *A. salmonicida* culture supernatant compared to their mock-treated counterparts. Elimination of NeuAc exposes cryptic structures, i.e., mainly Gal, GlcNAc and GalNAc, which did not significantly inhibit the *A. salmonicida* AI-2 secretion. This evidence supports the view that NeuAc is sensed by bacteria and reduces the synthesis of AI-2 signals. Furthermore, we have shown earlier that NeuAc is the terminal residue that primarily binds *A. salmonicida* [5], and terminal NeuAc shields the growth-promoting GlcNAc from *A. salmonicida* access [8]. Together, these results suggest a prominent role for NeuAc in *A. salmonicida* growth, behavior, and interactions with its host. In addition to being abundant on Atlantic salmon mucins, NeuAc is also abundant on mucin glycans from arctic charr [46].

We found the central player on mucins in reducing AI-2 secretion in *A. salmonicida* to be NeuAc. Mucins from all organ groups have NeuAc present terminally on *O*-glycans in high abundance and they reduce AI-2 secretion of *A. salmonicida* to a similar extent. Thus, the response of *A. salmonicida* to Atlantic salmon mucins does not seem to be organ-specific but rather a general response regardless of where on the body the first contact is made with the host. This can possibly reduce the resistance of the bacteria towards the immune system of the fish and reduce *aerB* virulence factor expression [12]. During infection, this effect of the mucins might help the host to prolong the invasion process and eliminate the bacteria by trapping them and flushing them out of the body or shedding them from the body surface [47] before the pathogen reaches the epithelial cells. It is likely that AI-2 inhibition by salmon mucins further supports the strong contact with *A. salmonicida*, as other bacterial species have been shown to have stronger effacing behavior with higher AI-2 signals [37,38,48]. When colonization occurs, this effect on AI-2 signals in *A. salmonicida* is accompanied by a variety of environmental factors, e.g., salinity, pH, and autoinducer molecules from other colonizing bacteria and the *A. salmonicida*-s AI-1 system further complicating the AI-2 machinery in an in vivo infection.

In conclusion, we showed that the bioluminescent-reporter assay with *V. harveyi* BB170 can be used to quantify AI-2 production with precautions against media components and that exogenous AI-2 signals increase the growth of *A. salmonicida.* Since AI-2 is a common bacterial language, this suggests that *A. salmonicida* might benefit from hijacking these signals from surrounding bacteria in competitive environments. We showed that Atlantic salmon mucins from the skin, pyloric ceca, proximal intestine and distal intestine reduce the AI-2 secretion of *A. salmonicida* in a NeuAc and *luxS* dependent manner. Together with previously published results [5,8], this suggests a prominent role for NeuAc in Atlantic salmon defense, regulating *A. salmonicida* growth, behavior and interactions with its host.

## 4. Materials and Methods

### 4.1. Bacterial Strains and Culture Conditions

*A. salmonicida* ssp. *salmonicida* strain VI-88/09/03175 (culture collection, Central Veterinary Laboratory, Oslo, Norway) was cultured in Brain Heart Infusion broth (BHI) at 19 °C and stocks were stored in BHI/glycerol 1:1 at −80 °C. S17-1 *E. coli* (listed in Table 1) was cultured in LB medium at 37 °C broth with vigorous agitation (250 rpm); when required, ampicillin or chloramphenicol was added to the medium to achieve a final concentration of 100 μg mL^−1^. *V. harveyi* strain BB170 was cultured overnight in Bioluminescence Autoinducer medium (ATCC medium: 2746) at 30 °C with continuous shaking at 100 rpm. Bacterial concentration was measured as OD_600_ nm using the spectrophotometer Genesys 30 (Thermo Scientific).

### 4.2. Culture Media

The DM medium contained 0.1 g/L Ca(NO_3_)_2_·4H_2_O, 0.4 g/L KCl, 0.1 g/L MgSO_4_·7H_2_O, 6 g/L NaCl, 0.8 g/L Na_2_HPO_4_, 2 g/L NaHCO_3_, 2 mg/L FeSO_4_, 5 g/L BSA, 50 mg/L adenine, 3 mg/L lipoic acid. For amino acids, DM medium contained 44.5 mg/L alanine, 632 mg/L arginine, 75 mg/L asparagine, 66.5 mg/L aspartic acid, 120 mg/L cysteine, 73.5 mg/L glutamic acid, 300 mg/L glutamine, 37.5 mg/L glycine, 110 mg/L histidine, 262.5 mg/L isoleucine, 262 mg/L leucine, 362.5 mg/L lysine, 75.5 mg/L methionine, 165 mg/L phenylalanine, 57.5 mg/L proline, 52.5 mg/L serine, 238 mg/L threonine, 51 mg/L tryptophan, 180 mg/L tyrosine, 234 mg/L valine. The vitamin content of DM medium was as follows: 0.2 mg/L D-biotin, 3 mg/L choline chloride, 1 mg/L folic acid, 35 mg/L myo-inositol, 1 mg/L niacinamide, 1 mg/L p-aminobenzoicacid, 1.25 mg/L D-pantothenicacid, 1 mg/L pyridoxinehydrochloride, 0.2 mg/L riboflavin, 1 mg/L thiamine hydrochloride, 5 µg/L vitamin B12 (cyanocobalamin). The CM9 medium contained 3 g/L KH_2_PO_4_, 12.8 g/L Na_2_HPO_4_, 0.5 g/L NaCl, 1.0 g/L NH_4_Cl, 0.24 g/L MgSO_4_, 11.1 mg/L CaCl_2_ and 40 g/L Bacto^TM^ Casamino acids (BD sciences). See Table 2 for a side by side comparison of the media. The Bacto^TM^ Casamino acids derive from hydrolysis of casein further supplemented with an undisclosed amount of growth factors, cystine, maltose, iron and inorganic salts, therefore making CM9 medium a nondefined medium. In addition, due to the high amount of carbohydrates, CM9 medium was not suitable for our assay to investigate the effect of mucins and saccharides on *A. salmonicida* extracellular AI-2 levels.

### 4.3. DNA Manipulation

Bacterial DNA-extraction manipulations were performed routinely, as described by Sambrook et al. [32]. All restriction enzymes were used according to the manufacturers’ instructions (New England Biolabs, Ipswich, MA, USA). PCR were carried out in an Applied Biosystems 2720 Thermal Cycler (Thermo Scientific) using Platinum™ II Hot-Start Green PCR Master Mix (2x) (Invitrogen). The reaction mixture (50 μL) contained 22 μL ddH_2_O, 25.0 μL Platinum™ II Hot-Start Green PCR Master Mix, 1.0 μL PCR forward primer (10 mM), 1.0 μL PCR reverse primer (10 mM), 1.0 μL DNA sample, using the following PCR program: initial denaturation for 2 min at 94 °C; 30 cycles each for denaturation for 15 s at 94 °C, annealing for 15 s at 60 °C; and extension for 1.5 min at 68 °C. DNA fragments for cloning purposes were extracted and purified from agarose gel using Qiagen Gel Extraction kit (Qiagen, Inc., Hilden, Germany). The homologous A and B fragments and pRE112 plasmid were connected by using Gibson Assembly^®^ Master Mix-Assembly (NEB Inc., Ipswich, MA, USA). *A. salmonicida* ssp. *Salmonicida* strain VI-88/09/03175 whole-genome sequencing was performed using the Illumina MiSeq platform v3 chemistry, 2 × 300 bp read length. Data QC, de novo assembly and annotation were performed using the BACTpipe pipeline [50]. From the annotated coding sequences, a S-ribosylhomocysteine lyase (*luxS*) homolog was identified, which was compared to its *A. salmonicida* ssp. *salmonicida* A449 reference strain equivalent using psi-BLAST (National Center for Biotechnology Information, NCBI).

### 4.4. Construction of ΔluxS Mutant

The *luxS* gene was knocked out with the suicide vector pRE112 (Table 3) by homologous recombination, according to the method described by Meng et al. [12]. Flanking regions of the *luxS* gene (A- upstream and B- downstream) were amplified from the *A. salmonicida* ssp. *salmonicida* strain VI-88/09/03175 gDNA template, using A-F/R and B-F/R primers and the plasmid pRE112 was digested by KpnI enzyme (NEB Inc., Ipswich, MA, USA) These two fragments were connected to the plasmid by using Gibson Assembly^®^ Master Mix-Assembly (NEB Inc., Ipswich, MA, USA). Subsequently, the plasmid ΔluxS_pRE112 was chemically transformed into *E. coli* S17-1 competent cells, containing *pir* and *sacB* gene, and chloramphenicol selection was performed for further bacterial conjugation with *A. salmonicida*. A sucrose selection (15%) for pRE112 absence was performed and confirmation of the ΔluxS genotype was made by PCR (Figure 7).

All the primers used in this study are shown in Table 3.

### 4.5. Quorum-Sensing Assay

For harvesting culture supernatant for AI-2 detection, *A. salmonicida* was cultured with 100 µg/mL mucin samples or with the same concentration of mono- and disaccharides, as described previously [8], in the denoted medium until cultures reached the stationary phase of growth. This is a time point where AI-2 production of *A. salmonicida* had reached maximum levels (Figure 2). Cultures were collected from wells and replicates were pooled and centrifuged at 4000× *g* for 5 min at RT. Supernatants were sterile-filtered (0.22 µm pore size, cellulose acetate filter, VWR), pipetted into white opaque 96-well plates and stored frozen at −20 °C ready for analysis. Freeze–thaw cycles and storage above −20 °C were avoided. Fresh, filtered *A. salmonicida* culture media (CM9 or DM) were used as negative controls in the bioreporter assay. To verify that the higher nutrient content in the negative control had no effect on the assay, we made a serial dilution from the negative control and subjected it to the bioreporter assay. No effect on nutrient dilution was detected in the DM media. Although a small effect could be detected in the CM9 media, the effect was less than 1% of the signal of the assays and therefore was considered negligible.

The measurement of autoinducer production was carried out with a quorum-sensing reporter bacterium: *Vibrio harveyi* transposon insertion mutant with the LuxN- phenotype (strain BB170 (ATCC^®^ BAA-1117™), courtesy of Bassler BL). *V. harveyi* strain BB170 was cultured overnight in Bioluminescence Autoinducer medium (ATCC medium: 2746) at 30 °C with continuous shaking at 100 rpm. The culture of the reporter strain having OD_600_ = 1.0 was diluted 1:5000 in fresh BA medium. 50 µL from the diluted *V. harveyi* BB170 suspension was added to 50 µL sterile-filtered *A. salmonicida* supernatant in each well of a white opaque 96-well cell-culture plate (Corning Inc., Corning, NY, USA). The mono- and disaccharides used in the study were tested separately and found to not interfere with the detection of AI-2 signals of the *A. salmonicida* supernatants. The plate was incubated at 30 °C with continuous shaking at 100 rpm and light production was measured in 10 min intervals for 12 h in a Clariostar plate reader. To evaluate the quantity of quorum-sensing signal molecules, one timepoint (corresponding to the induction phase) was used, where the decrease in light production turned into an increase. This corresponds to the lowest luminescence value of the “U”-shaped luminescence curve (Figure 10). The higher the concentration of AI-2 added, the lower the reduction in luminescence before the luminescence-autoinduction phase of *V. harveyi* BB170 began.

### 4.6. Assay Sensitivity Testing

The sensitivity of the *V. harveyi* BB170 bioluminescent assay was tested with 4,5-dihydroxy-2,3-pentanedione (DPD, Sigma Aldrich). The sensitivity testing was carried out as described above, except that instead of *A. salmonicida* supernatants, DPD diluted to the desired concentration in 50 µL DM and CM9 media was used.

### 4.7. Growth Assay

*A. salmonicida* cells, at a concentration equivalent to an optical density at 600 nm (OD_600_) of 0.1, were cultured in DM and CM9 media to test the linear range of detection of AI-2 signals and to analyze the effect of DPD on *A. salmonicida*. DM medium was used to test the effect of Atlantic salmon mucins and mucin saccharides on AI-2 production of *A. salmonicida*.

### 4.8. Growth Assay with DPD

DPD was diluted in 20 µL PBS and added to 80 µL *A. salmonicida* culture in 4–8 replicates of a Nunc-Delta surface flat-bottom plate (Nunc A/S, Roskilde, Denmark) to reach the specified final concentrations. 20 µL PBS added to 80 µL culture was used as a negative control. The assay was carried out at 23 °C and 120 rpm and monitored both by OD_560_ readings and fluorescent detection of AlamarBlue (Invitrogen) reduction. Changes in growth were calculated in the logarithmic phase of growth and were expressed relative to the negative control.

### 4.9. Growth Assay with Mucins and Saccharides

Purified mucin samples in 4 M GuHCl were dialyzed eight times against PBS and diluted in sterile PBS to a concentration of 100 µg/mL. The saccharides were used in the same concentration. The GuHCl-containing isolation buffer was dialyzed and diluted in parallel and used as a reference of normal growth (dialysis control [DC]). PBS was used as a reference for growth in the presence of monosaccharides and oligosaccharides. Monosaccharides were purchased from Sigma-Aldrich. GD1a-oligosaccharide, GlcNAcβ1,3Gal, and GalNacβ1,3Gal were purchased from Elicityl, France. Core 1 glycan (Galβ1,3GalNAc) was purchased from Dextra Laboratories, UK. Bacteria were cultured in 96-well Nunc-Delta surface flat-bottom plates (Nunc A/S, Roskilde, Denmark) in 4 to 8 replicates at 23 °C and 120 rpm in the presence of alamarBlue (Invitrogen) to monitor reduction of the dye. AlamarBlue reduction corresponds to the metabolism of the culture and correlates well with CFU/mL [8].

### 4.10. Fish and Sampling Procedure

Atlantic salmon parr (Långhult lax, Långhult, Sweden) were transported to the Department of Biology and Environmental Science and kept in 500 L tanks. The fish were held in recirculating 10 °C fresh water (FW), supplemented with 10% salt water (SW) (yielding a salinity of 2–3 ‰), at a flow rate of 8.5 l/min. The fish were exposed to a simulated natural photoperiod and hand-fed ad libitum once daily with a commercial dry pellet (Nutra Olympic 3 mm; Skretting Averøy, Ldt., Stavanger, Norway).

Five fish (31.06 ± 0.49 cm bl and 280.70 ± 12.78 g bw; mean ± Standard Error of the Mean) were randomly netted, anesthetized in metomidate (12.5 mg/L) and killed with a sharp blow to the head. Mucus from the skin was sampled by gently scraping the skin off, including mainly the epidermis and secreted mucus of the entire fish, using two microscopy glass slides. The fish was then opened longitudinally and the intestine, from the last pyloric ceca to the anus, was quickly dissected out. The intestine was cut open along the mesenteric border and the proximal region was separated from the distal at the ileorectal valve and the mucus and mucosa was scraped off using microscopy slides. The pyloric ceca were dissected out using small scissors and were pulverized in liquid nitrogen. All samples were placed in 10 mM sodium di-hydrogen phosphate containing 0.1 mM phenylmethanesulphonyl fluoride (PMSF), pH 6.5 (sampling buffer), to inhibit proteolytic cleavage. The experiment on fish in the present study was approved by the Ethical Committee for Animal Experiments in Gothenburg, Sweden under licence #46/2009. All methods involving fish were performed in accordance with the relevant guidelines and regulations.

### 4.11. Isolation and Purification of Mucins

The scrapings and pulverized tissues in sampling buffer were placed into five sample volumes of extraction buffer (6 M GuHCl, 5 mM EDTA, 10 mM sodium phosphate buffer, pH 6.5, containing 0.1 M PMSF), dispersed with Dounce homogenizer (four strokes with a loose pestle) and stirred slowly at 4 °C, overnight. The insoluble material was removed by centrifugation at 23,000× *g* for 50 min at 4 °C (Beckman JA-30 rotor) and the pellet was re-extracted twice with 10 mL extraction buffer. The supernatants from these three extractions were pooled and contained the GuHCl soluble mucins, hereafter used in our assays. The samples were filled up to 26 mL with extraction buffer. CsCl was added to the samples by gentle stirring and the samples were transferred to Quick Seal ultracentrifuge tubes (Beckman Coulter). The tubes were filled with 10 mM NaH_2_PO_4_ to give a starting density of 1.35 g/mL and samples were subjected to density-gradient centrifugation at 40,000× *g* for 90 h at 15 °C. The fractions were collected from the bottom of the tubes with a fraction collector equipped with a drop counter. Density-gradient fractions of purified mucin samples were analyzed for carbohydrates as periodate-oxidizable structures in a microtiter-based assay: Fractions diluted 1:100, 1:500 and 1:1000 in 4 M GuHCl were coated into 96-well plates (PolySorpTM, NUNC A/S, Roskilde, Denmark) and incubated overnight at 4 °C. The rest of the assay was carried out at 23–24 °C. After washing three times with washing solution (5 mM Tris-HCl, 0.15 M NaCl, 0.05% Tween 20, 0.02% NaN_3_, pH 7.75), the carbohydrates were oxidized by adding 25 mM sodium metaperiodate in 0.1 M sodium acetate buffer, pH 5.5 for 20 min. The plates were washed again and the wells were blocked with DELFIA blocking solution (50 mM Tris-HCl, 0.15 M NaCl, 90 mM CaCl_2_, 4 mM EDTA, 0.02% NaN_3_, 0.1% BSA, pH 7.75) for 1 h. After further washing steps, the samples were incubated for 1 h with 2.5 mM biotin hydrazide in 0.1 M sodium acetate buffer, pH 5.5, followed by another washing step. Europium-labeled streptavidin was diluted 1:1000 in Delfia assay buffer (50 mM Tris-HCl, 0.15 M NaCl, 20 mM DTPA, 0.01% Tween 20, 0.02% NaN_3_, 1.5% BSA, pH 7.75) and was added to the wells. After 1 h incubation, the plates were washed six times and then incubated with Delfia enhancement solution (0.05 M NaOH, 0.1 M ftalat, 0.1% Triton X-100, 50 mM TOPO, 15 mM b-NTA) for 5 min on an orbital shaker. Fluorescence (λ_excitation_ = 340 and λ_emission_ = 615) was measured using a Wallac 1420 VICTOR_2_ plate reader with the Europium label protocol (PerkinElmer, Waltham, MA, USA). Density measurements were performed using a Carlsberg pipette as a pycnometer: 300 µL of sample was aspirated into the pipette and weighed, and density was calculated as g/mL. DNA content was measured by UV light absorbance at 260 nm.

### 4.12. Preparation of Mucin Samples

Gradient fractions containing mucins were pooled together to obtain one sample for each gradient. Mucin concentrations in pooled samples were determined by serial dilution of the samples, as well as a standard curve of a fusion protein of the mucin MUC1, 16TR and IgG2a Fc starting at a concentration of 20 mg/mL and using seven 1:2 serial dilutions and detection of carbohydrate as periodate-oxidizable structures in a microtiter-based assay, as described above. The mucin concentrations were calculated from the standard curve. This method of concentration determination was chosen as not all mucins come into solution after freeze drying, and determining concentration by freeze drying can therefore contain large errors or remove mucin species selectively. Although this is not an exact measure of concentration, it can be used to ensure that the mucins are at the same concentration for comparative assays, and since bacteria–mucin interactions largely occur via the mucin glycans [6], setting the concentration based on the glycan content appears most appropriate.

### 4.13. Statistical Analyses

Statistical analyses were performed using the GraphPad Prism 7.0 software package (GraphPad Software Inc., London, UK). Student *t*-tests and one-way analysis of variance (ANOVA) followed by Dunnett’s post hoc test were used to compare groups and treatments, as indicated in the figure legends. The relationship between DPD and *A. salmonicida* growth was tested with two-tailed Pearson correlation test. The level of significance was set at *p* ≤ 0.05.

## Figures and Tables

**Figure 1 ijms-23-04326-f001:**
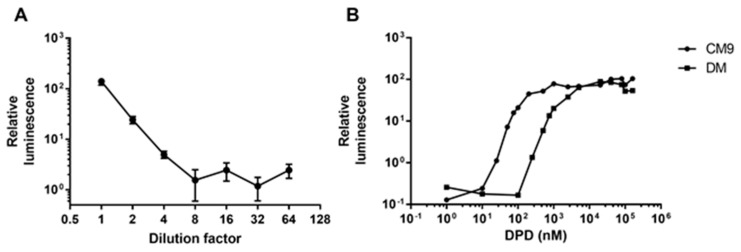
Media-dependent sensitivity of the *V. harveyi* BB170 autoinducer-2 assay. *V. harveyi* BB170 luminescence was measured in the QS reporter assay in response to addition of *A. salmonicida* cell-free culture supernatant (**A**) or DPD in DM or CM9 media (**B**). (**A**) Up to four-fold dilution of *Vibrio harveyi* BB170 cell-free culture supernatant has a linear relationship with relative luminescence of the AI assay (r^2^ = 0.699). (**B**) In CM9 medium, quantification is reliable in the range of 10 nM to 100 nM DPD molecules (r^2^ = 0.98), while in DM medium the range is between 100 nM and 1 µM (r^2^ = 0.98). Relative luminescence values were calculated by normalizing absolute luminescence with the negative control. Data points are expressed as Mean ± Standard Error of the Mean (*n* = 4). The results were reproduced three times and a representative experiment is shown here. Abbreviations: DPD = 4,5-dihydroxy-2,3-pentanedione.

**Figure 2 ijms-23-04326-f002:**
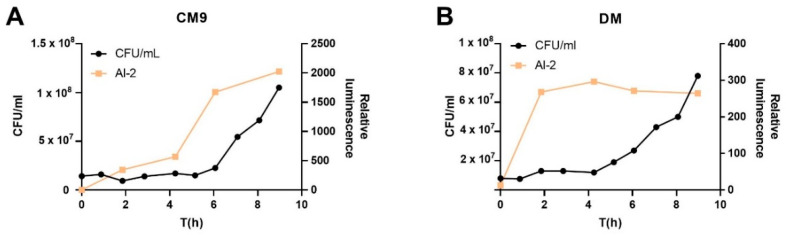
Autoinducer-2 production in *A. salmonicida* cultures. Time course of AI-2 production by *A. salmonicida* cultured in CM9 (**A**) and DM (**B**).

**Figure 3 ijms-23-04326-f003:**
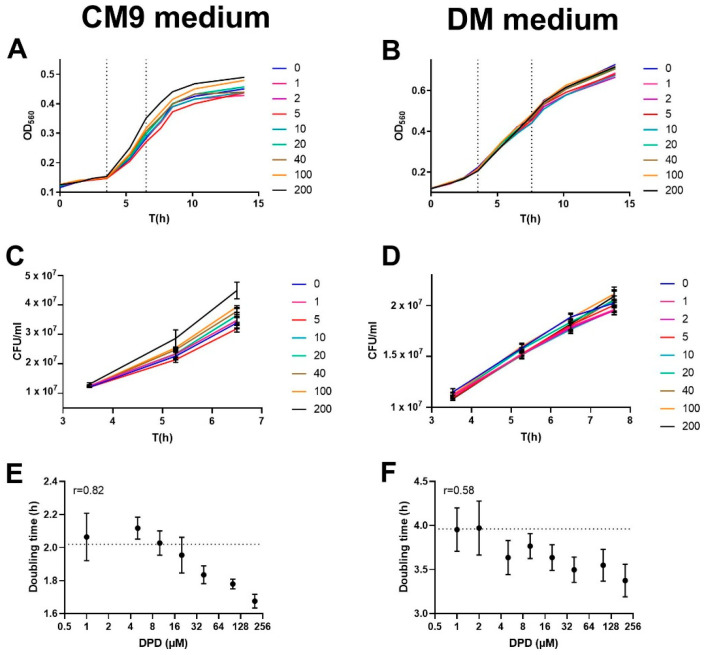
Effect of exogenous DPD on *A. salmonicida* growth. The effect of DPD on *A. salmonicida* growth was monitored over time (**A**,**B**) and correlated against *A. salmonicida* doubling time (**E**) ( *p* = 0.0002; r^2^ = 0.82 in CM9 and (**F**); *p* = 0.017; r^2^ = 0.58 in in DM). *A. salmonicida* growth curves in the log phase of growth expressed as CFU/ml in CM9 (**C**) and DM (**D**) media. The horizontal dotted lines in panels (**E**,**F**) denote the doubling time of the negative control (no added DPD). Data points are expressed as Mean ± Standard Error of the Mean (*n* = 4). Statistical test: Pearson correlation test. Abbreviations: DPD = (S)-4,5-Dihydroxy-2,3-pentandione. The results are based on three independent experiments.

**Figure 4 ijms-23-04326-f004:**
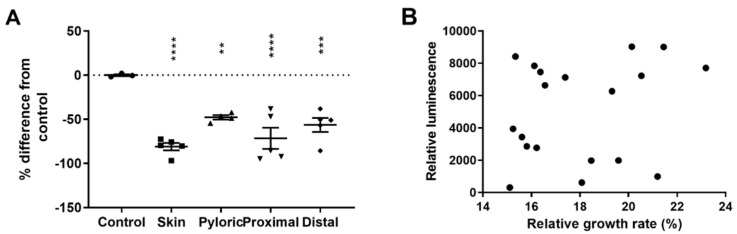
Effect of salmon mucins on *A. salmonicida* AI-2 levels in the culture medium. Extracellular AI-2 levels were measured after culturing *A. salmonicida* to the late log phase in the presence of 100 µg/mL Atlantic salmon mucins. These experiments were performed in DM medium to avoid interference of saccharides present in the CM9 medium. Mucins from all organ sites decreased the AI-2 levels in the *A. salmonicida* supernatant (skin, pyloric ceca, proximal intestine and distal intestine *p* = 0.0001, *p* = 0.0049, *p* = 0.0001 and *p* = 0.0008, respectively; (**A**). Data points correspond to biological replicates of individual fish and are expressed as Mean ± Standard Error of the Mean (*n* = 5). Statistical test: One-way ANOVA with Dunnett’s post hoc test. ** = *p* ≤ 0.01; *** = *p* ≤ 0.001; **** = *p* ≤ 0.0001. AI-2 production and growth-modulating effect of mucins did not correlate (r^2^ = 0.05, *n* = 19, *p* = *n*.s.); (**B**). Statistical test: two-tailed Pearson correlation test. Abbreviations: Pyloric = Pyloric ceca; Proximal = proximal intestine and Distal = distal intestine. The results were reproduced three times with similar results and the assay was performed within its linear range.

**Figure 5 ijms-23-04326-f005:**
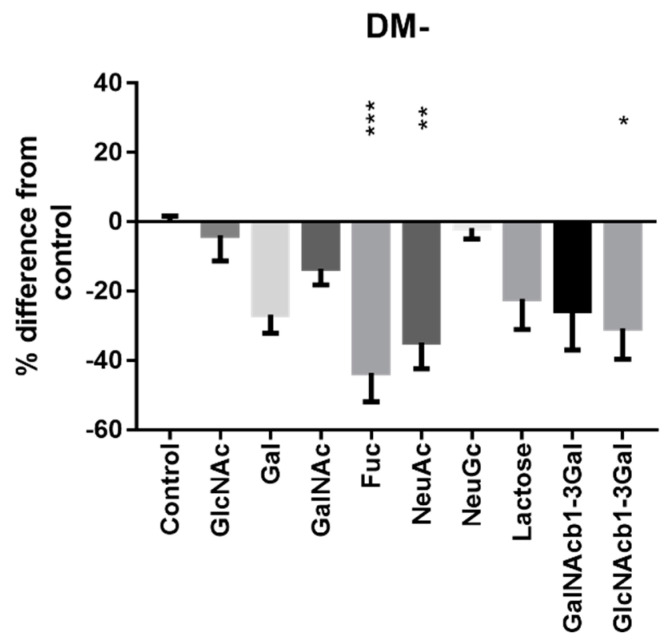
Effect of saccharides on *A. salmonicida* AI-2 levels. Mono- and disaccharides in 100 µg/mL concentration were added to *A. salmonicida* cultures in DM medium, after which extracellular AI-2 levels were measured from the late log phase. Fucose, NeuAc and GlcNAcβ1,3Gal decreased the AI-2 secretion of *A. salmonicida* in DM medium (*p* = 0.0008; *p* = 0.0101 and *p* = 0.0297, respectively). Data points are expressed as Mean ± Standard Error of the Mean (*n* = 6) Statistical test: One-way ANOVA with Dunnett´s post hoc test. * = *p* ≤ 0.05, ** = *p* ≤ 0.01, *** = *p* ≤ 0.001. The results are based on two independent experiments and the graph includes all data points from the two experiments. The data points are normalized for *A. salmonicida* growth, and the assay was performed within its linear range. Abbreviations: Glc = glucose; GlcNAc = *N*-acetylglucosamine; Gal = galactose; GalNAc = *N*-acetylgalactosamine; Fuc = fucose; NeuAc = *N*-acetylneuraminic acid; NeuGc = *N*-glycolylneuraminic acid.

**Figure 6 ijms-23-04326-f006:**
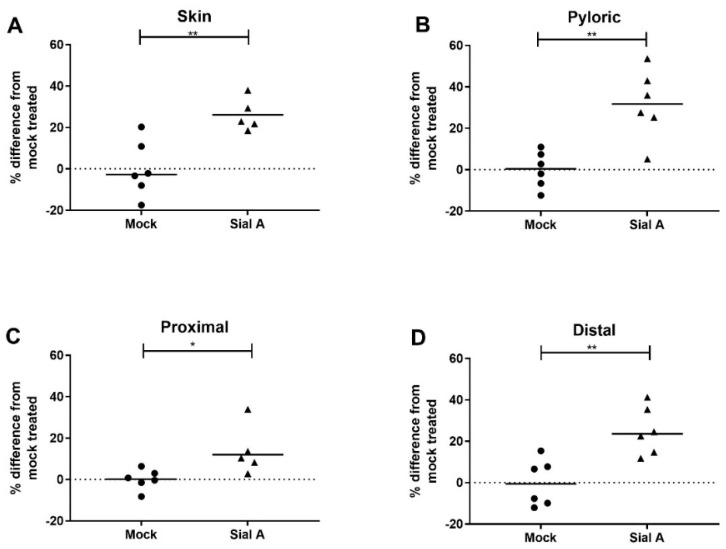
Effect of sialidase treatment on the effect of mucins on *A. salmonicida* AI-2 secretion. Extracellular AI-2 was measured after *A. salmonicida* culture with Atlantic salmon mucins treated with Sialidase A or with the same treatment in the absence of the enzyme (mock). All modified mucins from the four groups increased the AI-2 quantity relative to their mock-treated counterparts (**A**: skin, **B**: pyloric ceca, **C**: proximal intestine and **D**: distal intestine *p* = 0.0041, *p* = 0.0021, *p* = 0416 and *p* = 0.0035, respectively; *n* = 6). Statistical test: Student´s *t*-test. * = *p* ≤ 0.05, ** = *p* ≤ 0.01. The data points are normalized for *A. salmonicida* growth. Abbreviations: Mock = enzyme buffer-treated negative-control mucins; Sial A = sialidase A-treated mucins; pyloric = pyloric cecal mucins.

**Figure 7 ijms-23-04326-f007:**
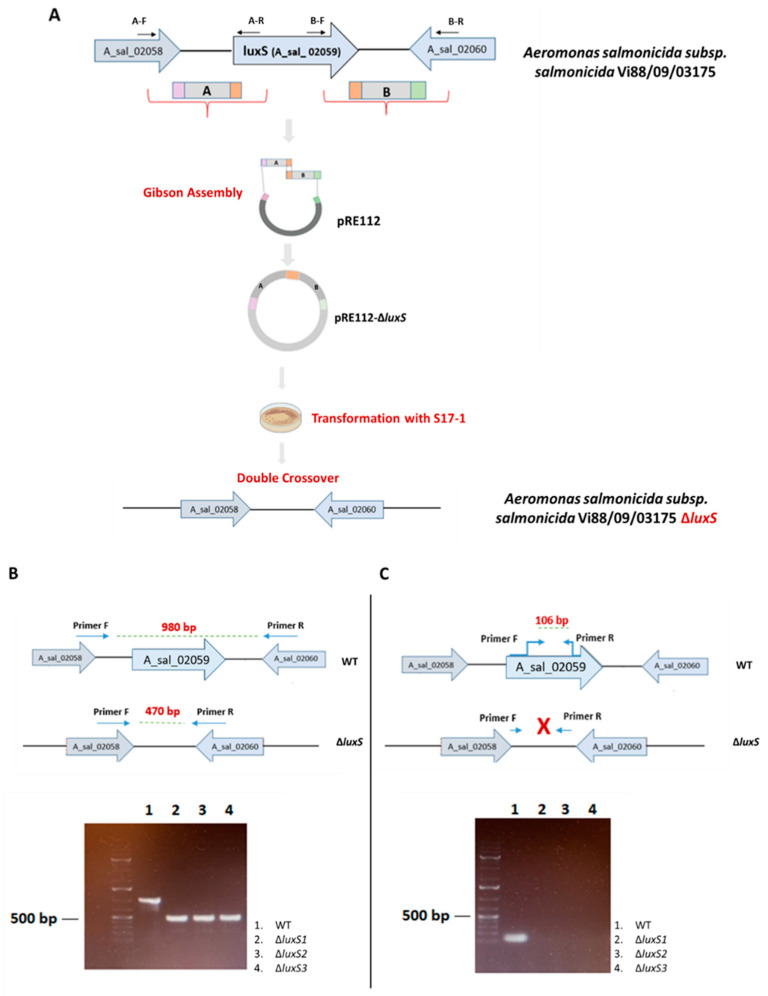
Construction of *A. salmonicida* ssp. *salmonicida* strain VI-88/09/03175 ΔluxS mutant. (**A**) Deletion of the luxS gene by overlapping PCR using Gibson Assembly (NEB Inc., Ipswich, MA, the USA). (**B**,**C**) The deletion was verified by PCR using genomic DNA as template with two approaches: (**B**). Gel electrophoresis for PCR with primers amplifying flanking regions of luxS gene; the WT shows a band with 980 bp while the ΔluxS mutant clones (2,3,4) show bands with a size of 470 bp, in accordance with the theoretical size. (**C**). Gel electrophoresis confirming deletion of the gene using primers mapping inside the luxS gene; the WT shows a band with 106 bp while the ΔluxS mutant clones (2,3,4) shows no band, indicating deletion of the gene. A 1 kb plus DNA ladder (Thermo Scientific, Whaltham, MA, USA) was used as molecular-weight marker.

**Figure 8 ijms-23-04326-f008:**
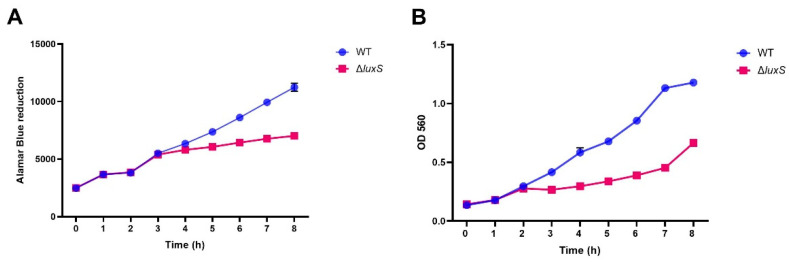
Growth comparison of the *A. salmonicida* ssp. *salmonicida* strain VI-88/09/03175 WT strain and its ΔluxS mutant. *A. salmonicida* WT and ΔluxS were cultured in DM with a starting optical density at 600 nm of 0.1. The assay was carried out at 20 °C and 120 rpm and monitored both by fluorescent detection of AlamarBlue reduction (**A**: *n* = 6) and OD_560_ (**B**: *n* = 4) readings every hour. Data points are expressed as Mean ± Standard Error of the Mean.

**Figure 9 ijms-23-04326-f009:**
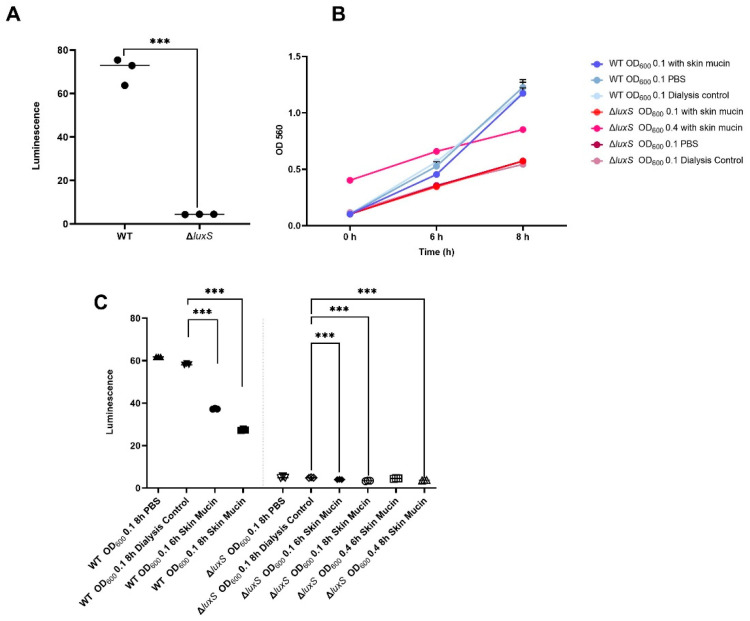
AI-2 production in *A. salmonicida* ssp. *salmonicida* strain VI-88/09/03175 WT and VI-88/09/03175ΔluxS in the presence and absence of Atlantic salmon skin mucins. (**A**) AI-2 activity in cell-free media from *A. salmonicida* WT and the ΔluxS mutant after 8 h of culture in DM with a start OD_600_ of 0.1. Data points are expressed as Mean ± Standard Error of the Mean (*n* = 3). (**B**) *A. salmonicida* wt and ΔluxS were cultured in DM and growth measured at OD_560_. Data points are expressed as Mean ± Standard Error of the Mean (*n* = 5). (**C**) AI-2 activity in cell-free media from *A. salmonicida* WT and the ΔluxS mutant in the presence and absence of Atlantic salmon skin mucins. In the dialysis control sample, mucin extraction buffer dialyzed against PBS in parallel to the mucin samples was added instead of mucins in PBS to the DM prior to addition of the *A. salmonicida*. That the AI-2 levels are similar in these cultures as in the samples where PBS was added suggest that no chemicals that affect the AI-2 activity remain from the mucin-isolation process. In order to investigate if the difference in cell density caused by differences in growth rate between the ΔluxS and WT *A. salmonicida* interfered with the assay, we also measured AI-2 activity after 6 and 8 h and included cultures with a starting OD_600_ of 0.4 starting for the ΔluxS in addition to the standard starting OD_600_ of 0.1, and the level of AI-2 activity in these samples were relatively similar to those with the same treatment at 8 h. Statistics: ANOVA with Dunnet´s post hoc test, *n* = 3, *** = *p* ≤ 0.001.

**Figure 10 ijms-23-04326-f010:**
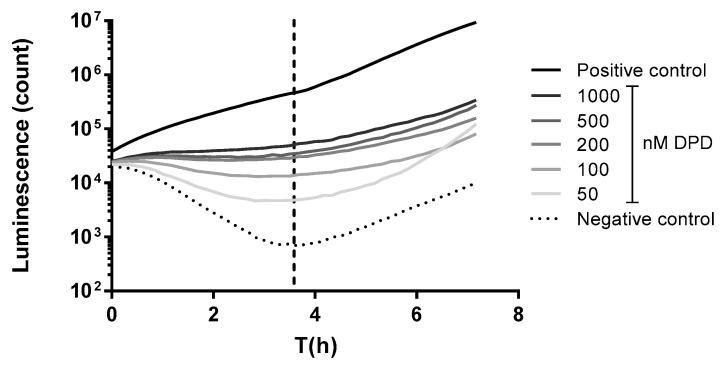
Light-production time course of *V. harveyi* BB170 reporter. This graph is an example showing the timepoint selection for evaluation of luminescent signals for standard curves and subsequent calculation of DPD equivalent AI-2 values. *V. harveyi* BB170 luminescence was measured in the QS-reporter assay in response to CM9 medium (negative control), DPD dissolved in CM9 medium and *V. harveyi* own supernatant (positive control). The dashed vertical line shows the time point corresponding to the start of autoinduction in *V. harveyi* coinciding with the lowest luminescence count of the negative control. Data points in the intersection of curves and the dashed line were used to plot “DPD concentration-relative luminescence” standard curves. Data points are expressed as Means (*n* = 3). Abbreviations: DPD = 4,5-dihydroxy-2,3-pentanedione.

**Table 1 ijms-23-04326-t001:** Strains and plasmids in this study.

Strain or Plasmid	Description	Source/Reference
*A. salmonicida* ssp. *salmonicida* strain VI-88/09/03175	-	Norwegian Veterinary Institute, Norway
*A. salmonicida* ssp. *salmonicida* strain VI-88/09/03175 Δ*luxS*	*luxS* deficient	This study
*E. coli* S17-1	Mobilizing donor for conjugation RP4-2(Km::Tn7,Tc::Mu-1), pro-82, LAMpir, recA1, endA1, thiE1, hsdR17, creC510	[49]
*V. harvey* BB170	Reporter for AI-2 bioassay	ATCC BAA-1117
pRE112 plasmid	Suicide vector; R6K ori sacB Cm^R^	Addgene
ΔluxS_pRE112 plasmid	Cmr; pRE112 bearing homologous arms of luxS	This study.

**Table 2 ijms-23-04326-t002:** Comparison of the two media side-by-side.

Component	DM Medium	CM9 Medium
Ca(NO_3_)_2_·4H_2_O	0.1 g/L	
KCl	0.4 g/L	
MgSO_4_·7H_2_O	0.1 g/L	
NaCl	6 g/L	0.5 g/L
Na_2_HPO_4_	0.8 g/L	12.8 g/L
NaHCO_3_	2 g/L	
FeSO_4_	2 mg/L	
BSA	5 g/L	
adenine	50 mg/L	
lipoic acid	3 mg/L	
alanine	44.5 mg/L	
arginine	632 mg/L	
asparagine	75 mg/L	
aspartic acid	66.5 mg/L	
cysteine	120 mg/L	
glutamic acid	73.5 mg/L	
glutamine	300 mg/L	
glycine	37.5 mg/L	
histidine	110 mg/L	
isoleucine	262.5 mg/L	
leucine	262 mg/L	
lysine	362.5 mg/L	
methionine	75.5 mg/L	
phenylalanine	165 mg/L	
proline	57.5 mg/L	
serine	52.5 mg/L	
threonine	238 mg/L	
tryptophan	51 mg/L	
tyrosine	180 mg/L	
valine	234 mg/L	
D-biotin	0.2 mg/L	
choline chloride	3 mg/L	
folic acid	1 mg/L	
myo-inositol	35 mg/L	
niacinamide	1 mg/L	
p-aminobenzoicacid	1 mg/L	
D-pantothenicacid	1.25 mg/L	
pyridoxinehydrochloride	1 mg/L	
riboflavin	0.2 mg/L	
thiamine hydrochloride	1 mg/L	
vitamin B12 (cyanocobalamin)	5 µg/L	
KH_2_PO_4_		3 g/L
NH_4_Cl		1.0 g/L
MgSO_4_		0.24 g/L
CaCl_2_		11.1 mg/L
Bacto^TM^ Casamino acids		40 g/L

**Table 3 ijms-23-04326-t003:** Primers used in this study.

Primer	Sequence (5′−3′)
A-F	ctcgatatcgcatgcggtaccGAAGCCATAATCAAACCGTT
A-R	catgcacaacTTCTGACTCCTGATTTGGTTAC
B-F	ggagtcagaaGTTGTGCATGGCAAAAATGA
B-R	caagcttcttctagaggtaccGGATAACTATACCCTTTGGTATAAC
luxS-up-F	CTGAACGGCAATGGTGTGGAGA
luxS-down-R	GCACGGTCAACACATCGCTCAT
LuxS-F	ATGAGCGATGTGTTGACCGT
LuxS-R	CGATCTCATGGGCTTCCTCC

## Data Availability

The draft genome of *A salmonicida* ssp. *salmonicida* strain VI-88/09/03175 was submitted to GenBank under BioProject number PRJNA698804.

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
