# Peer review of "Atlantic Salmon Mucins Inhibit LuxS-Dependent A. Salmonicida AI-2 Quorum Sensing in an N-Acetylneuraminic Acid-Dependent Manner"

_ijms, 2022, doi:10.3390/ijms23084326_

Round 1

Reviewer 1 Report

Quorum sensing is the cell-cell communication between bacteria that allows the perception of population density by small signaling molecules and modifies gene expression in response to the population density. It controls a wide spectrum of processes and phenotypic behaviors, including stress resistance, biofilm formation, production of toxins and production of secondary metabolites. In addition to playing important roles in intraspecies and interspecies communication, quorum sensing is also involved in host-microbe interactions. Identifying a novel factors with anti-quorum sensing properties is important in aquaculture. Disease outbreaks limit development of the industry because of associated ethical and economic issues. The authors showed that Atlantic salmon mucins from the skin, pyloric ceca, proximal intestine and distal intestine reduce the AI-2 secretion of quorum sensing system in Aeromonas salmonicida in a NeuAc and luxS dependent manner. The results directly suggest a role for NeuAc in salmon defense, regulating the microbial growth, activity and interaction with its host. The researches have been properly planned and consistently implemented. Minor mistakes do not diminish the substantive value of the work.

Minor points:

  1. The abstract section should be enriched with the scope of the experiments and principal conclusions
  2. The introduction section should provide the context and sufficient background information for the study. In its current form, there are no links between the individual sets of information collected in paragraphs.
  3. The names of the microorganisms should be italicized

Author Response

We thank the reviewers for suggestions on how to improve our manuscript. We have addressed the issues raised by the reviewers below and highlighted the changes in the manuscript.

Sincerely

Sara Lindén

Reviewer 1

Quorum sensing is the cell-cell communication between bacteria that allows the perception of population density by small signaling molecules and modifies gene expression in response to the population density. It controls a wide spectrum of processes and phenotypic behaviors, including stress resistance, biofilm formation, production of toxins and production of secondary metabolites. In addition to playing important roles in intraspecies and interspecies communication, quorum sensing is also involved in host-microbe interactions. Identifying a novel factors with anti-quorum sensing properties is important in aquaculture. Disease outbreaks limit development of the industry because of associated ethical and economic issues. The authors showed that Atlantic salmon mucins from the skin, pyloric ceca, proximal intestine and distal intestine reduce the AI-2 secretion of quorum sensing system in Aeromonas salmonicida in a NeuAc and luxS dependent manner. The results directly suggest a role for NeuAc in salmon defense, regulating the microbial growth, activity and interaction with its host. The researches have been properly planned and consistently implemented. Minor mistakes do not diminish the substantive value of the work.

Minor points:

  1. The abstract section should be enriched with the scope of the experiments and principal conclusions

Response: We have revised the abstract, although since the maximum length of the abstract is 200 words, we were not able to add much.  However, the scope of the experiments and conclusions are present in the abstract.

  1. The introduction section should provide the context and sufficient background information for the study. In its current form, there are no links between the individual sets of information collected in paragraphs.

Response: We have added a connecting sentence between the first and second paragraph. The remainder of the paragraphs are already closely related.

  1. The names of the microorganisms should be italicized

Response: We have worked through the manuscript and italicized the names of the organisms where the formatting had been lost (results section).

Reviewer 2 Report

Thia is a very intersting on QS to A. salmonicida.

Author Response

Response: Thank you.

Reviewer 3 Report

This is a well planned and interesting study with linear flow of results and illustrating a possible mechanism.

Few comments

The abstract needs to be modified to get the flow. It seems it is written in bullet points.

Authors should show individual data points for the replicates on graphs.

Author Response

We thank the reviewers for suggestions on how to improve our manuscript. We have addressed the issues raised by the reviewers below and highlighted the changes in the manuscript.

Sincerely

Sara Lindén

Reviewer 3

This is a well planned and interesting study with linear flow of results and illustrating a possible mechanism.

Few comments

The abstract needs to be modified to get the flow. It seems it is written in bullet points.

Response: We have revised the abstract to improv the flow, although since the maximum length of the abstract is 200 words, we were not able to extend it.  

Authors should show individual data points for the replicates on graphs.

Response: we have re-plotted the graphs to show the individual datapoints when it was suitable to do this (fig 6 and 9). There are more figs that show median/mean with error bars, however, in these graphs, showing all individual datapoints obscured the results, and the mean/median with error bars resulted in more clear graphs, so we kept the original format. Presenting data as mean or median with error bars is a valid and commonly used format for presenting data, especially if it improves the visualizations of the results.